# Beyond Soil Inoculation: Cyanobacteria as a Fertilizer Replacement

Michael S. Massey [1] and Jessica G. Davis [2,*]

1 Environment Canterbury, Christchurch 8140, New Zealand; michael.massey@ecan.govt.nz
2 Agricultural Experiment Station, Colorado State University, Fort Collins, CO 80523, USA
* Correspondence: jessica.davis@colostate.edu

**Abstract:** Nitrogen-fixing bacteria such as cyanobacteria have the capability to fix atmospheric nitrogen at ambient temperature and pressure, and intensive cultivation of cyanobacteria for fertilizer could lead to its use as an "environmentally friendly" replacement or supplement for nitrogen (N) fertilizer derived from the Haber–Bosch process. Prior research has focused on the use of N-fixing bacteria as a soil inoculum, and while this can improve crop yields, yield improvements are generally attributed to plant-growth-promoting substances produced by the bacteria, rather than to biological N fixation. The intensive cultivation of cyanobacteria in raceways or bioreactors can result in a fertilizer that provides N and organic carbon, as well as potentially similar growth-promoting substances observed in prior research work. On-farm or local production of cyanobacterial fertilizer could also circumvent infrastructure limitations, economic and geopolitical issues, and challenges in distribution and transport related to Haber–Bosch-derived N fertilizers. The use of cyanobacterial N fertilizer could have many agronomic and environmental advantages over N fertilizer derived from the Haber–Bosch process, but study of cyanobacteria as a replacement for other N fertilizers remains very limited. Scientific and practical challenges remain for this promising but as-yet unproven approach to maintaining or improving soil N fertility.

**Keywords:** biological N fixation; Haber–Bosch process; cyanobacteria

## 1. Going beyond Soil Inoculation

Nitrogen (N) is the major nutrient needed by crops in the largest quantities. Sources of N fertilizer have historically included mined Chilean nitrate ($NaNO_3$), guano from birds and bats, and farmyard manure. After the Haber–Bosch process was developed in the early 1900s to convert $N_2$ from the atmosphere into $NH_3$ (gas), $NH_3$-based fertilizers have dominated the market, including urea, ammonium nitrate, and mixed fertilizers, such as mono- or di-ammonium phosphates. The global human population has increased from fewer than 2 billion people in the early 1900s to more than 8 billion people today, magnifying the demand for N fertilizer to grow crops and raise livestock for human consumption.

Cyanobacteria have long been recognized to occur naturally in flooded rice paddies and to benefit rice through biological N fixation [1]. The inoculation of soil with N-fixing bacteria has been considered as a method of increasing N availability for dryland crops, not only those grown in paddies. In particular, heterotrophic N-fixing bacteria such as *Azospirillum* sp. have been used as an inoculum [2,3] and consume organic matter in the soil for their growth, as well as to power the fixation of "reactive" nitrogen (e.g., ammonium or nitrate) from atmospheric $N_2$ with the nitrogenase enzyme. Inoculation of soil with heterotrophic N-fixing bacteria can increase yields [2,4], but yield increases were attributed to promotion of root growth, rather than to biological N fixation [4]. More recent research supports the finding that inoculation with *Azospirillum* sp. can improve plant growth and increase resilience, even if its use as a N fertilizer is limited [5]. Cyanobacteria are found in soil crusts and have been studied for their potential to improve the physical properties of

soil, in addition to potential chemical changes [6–8]. Others investigated the inoculation of soil with cyanobacteria as a fertilizer [9–11], but the inoculum could not compete with native microorganisms [9]. In a pot study, it was reported that cyanobacterial inoculation enhanced soil inorganic N by less than 3 mg kg$^{-1}$, a very small amount compared to typical crop N uptake rates of 50–100 mg kg$^{-1}$ or more [12]. In other research, inoculation with cyanobacteria improved rice yields, but issues with contamination, growing season limitations, and slow growth rate reportedly made the practice unpopular with farmers [1]. Soil can also be inoculated with other microorganisms (e.g., mycorrhizal fungi such as Glomeromycetes, Ascomycetes, or Basidiomycetes) which can enhance plant N uptake, but which do not themselves appreciably increase the quantity of soil N available for plant uptake.

Symbiotic N fixation (e.g., with leguminous plants and N-fixing *Rhizobium* sp.) can, in contrast, provide for a plant's entire N requirement, and can substantially enhance soil N fertility. However, symbiotic N fixation comes at a cost in both time and water use and might, therefore, be impractical in situations where water is scarce or growing seasons are limited. The focus of this work is to look beyond the inoculation of soil, and beyond symbiotic N fixation, to the use of intensively grown autotrophic N-fixing bacteria (i.e., cyanobacteria) as a replacement for other, more common N fertilizers such as urea. Objectives of this review are to explain some of the disadvantages of N fertilizers produced through the Haber–Bosch process, to elucidate the potential benefits of cyanobacterial fertilizer, and to explore the challenges for research needed to bring this opportunity to fruition.

## 2. Disadvantages of Fertilizers Produced by the Haber–Bosch Process

High-temperature (circa 500 °C), high-pressure (circa 300 atm) fixation of ammonia ($NH_3$) from atmospheric $N_2$ via the Haber–Bosch process provides the crucial feedstock for production of other common N fertilizers such as urea. Although almost 40% of the human population depends upon food grown using fertilizer produced through the Haber–Bosch process [13], on the other hand, the amount of anthropogenic N in the Earth system is roughly equal to the amount of natural N—that is, humans have doubled the amount of reactive N in the Earth system since the advent of the Haber–Bosch process [14–16]. This doubling of N pools and fluxes has come with environmental consequences [17], not the least of which is the amount of energy consumed by the process, in addition to energy used in distribution and transport. Depending on the country, fertilizer production, distribution, and application can account for a large proportion of energy use in modern diets. For example, Sandström et al. [18] found that fertilizer production and use accounts for about 14% of the total carbon footprint of the diets of people in the European Union. Process energy for urea, i.e., the energy used in the urea production process, amounted to circa 5.3 GJ/metric ton in 2019; carbon dioxide emissions from N fixation amount to circa 2.0 t $CO_2$ per ton of fixed $NH_3$ [19,20]; yet process energy reportedly only accounted for 10% of the total energy used for N fertilizer in agriculture [19]. Between manufacturing and distribution, energy usage and $CO_2$ emissions are major drawbacks to the fertilizers produced by the Haber–Bosch process. Once fertilizer is applied it can drive further $CO_2$ emissions from increased mineralization of soil organic matter [21], as well as emission of nitrogenous greenhouse gases such as nitrous oxide [22].

There are also economic and geopolitical implications, as well as infrastructure limitations to the manufacture and distribution of fertilizers produced by the Haber–Bosch process. Methane (natural gas) is the most common source of hydrogen (H) for the production of $NH_3$ from $N_2$ gas, and natural gas can also be used as an energy source for N fixation in the Haber–Bosch process. When natural gas prices rise, such as when supplies are restricted due to geopolitical instability, fertilizer prices also tend to rise, with wide-ranging impacts. Nitrogen fixation requires a reliable supply of H, as well as power, so cannot occur in places where adequate supplies are not available. Therefore, the product must be distributed after manufacture, and the point of use might be very remote from the point of manufacture. Distribution requires infrastructure which might not be avail-

able at all in some areas, or if infrastructure is available, the costs of transport could be considerably higher than in industrialized nations with extensive transportation networks. Infrastructure limitations contribute to high fertilizer prices, limited fertilizer availability, and limited fertilizer use in some areas of the world. Indeed, these factors contribute to observed imbalances in agricultural development, with some areas facing persistent soil nutrient deficits, while other regions have substantial soil nutrient surpluses [23] which, in turn, result in negative impacts to water quality and human and ecological health [17].

Recent research efforts have focused on improvements to the Haber–Bosch process, such as the development of new catalysts for lower-temperature N fixation [24,25]. Additionally, progress has been made on moving away from fossil fuel use for N fixation, often also with a focus on the use of new catalysts or renewable energy sources [20,26], or electrochemical reduction of $N_2$ to $NH_3$ [27]. While these approaches are promising, they only address issues with the production of common fertilizers such as urea—while not addressing difficulties in distribution, or issues that occur after the fertilizer is applied to the soil. In contrast, distributed production of biologically based N fertilizer with some organic matter content has the potential to at least partially address some of the drawbacks of fertilizers made using $NH_3$ from the Haber–Bosch process.

## 3. Potential Benefits of Cyanobacterial Fertilizer

Cyanobacteria are a diverse phylum of Gram-negative bacteria that exhibit different morphological characteristics (single-celled, filamentous, etc.) and inhabit many environments (terrestrial, freshwater aquatic, marine, etc.), though they might be dormant in certain conditions or at certain times. Cyanobacteria can photosynthesize, and some can use the energy gained from photosynthesis to fix inert $N_2$ gas from the atmosphere into biologically reactive forms of N such as $NH_3$. Unlike the Haber–Bosch process, cyanobacterial N fixation occurs at ambient temperature and pressure. In cyanobacteria, photosynthesis and N fixation are separated either in space (for example, through specialized cells where N fixation takes place) or in time (daytime photosynthesis and nighttime N fixation) within individual bacterium. By intensively growing and harnessing cyanobacteria as a fertilizer, we can effectively use solar power through photosynthesis to fix N into plant-available forms and decrease or eliminate the use of fossil fuels for N fertilizer production. Cyanobacteria production methods can be applied across a variety of scales, from a large-scale industrial facility to small-scale, on-farm production. Therefore, a distributed model where farmers or farming communities grow their own fertilizer should be explored (Figure 1). If cyanobacterial fertilizer was grown on-farm or within the local community, the use of fossil fuels for distribution and transport, and the associated costs, could be greatly reduced.

Photosynthesis also effectively sequesters $CO_2$ [28], and additional C from application of live cyanobacteria to land often results in increases in microbial biomass, while also improving soil health parameters such as aggregate stability and water retention [29]. Along with decreased $CO_2$ emissions, improvements in $CO_2$ sequestration in soil might therefore result from application of cyanobacteria to soil as fertilizer.

On-farm cyanobacterial fertilizer production offers flexibility in both its production and use. For example, it can be produced as certified organic fertilizer [30] or using conventional sources of nutrients [31]. In addition, it can be utilized directly and applied as a liquid as described above, or it can be dried down and applied as a solid fertilizer [32,33]. When aeration and mixing is stopped, cyanobacterial biomass sinks in the production ponds, the supernatant can be pumped off, and the process is repeated until there is little remaining liquid. Then fans can be utilized to air-dry the cyanobacterial biomass for use as a solid fertilizer.

The N fixation process that takes place in cyanobacterial cells is facilitated by the nitrogenase enzyme and results in the production of amino acids (e.g., glutamine, glutamic acid). Since plants generally take up nitrate and ammonium, the amino acids must be broken down through the mineralization process to be converted into these inorganic forms. A 140 d laboratory incubation was used to compare the N mineralization of solid

cyanobacterial biofertilizer with compost [34]. The N availability (N in the nitrate and ammonium forms) from cyanobacterial biofertilizer was 31.6% at the end of the incubation, while compost demonstrated only 15.5% of the applied N to be plant-available in the same period of time. The incubation study was followed by a greenhouse study which showed that cyanobacterial biofertilizer resulted in a significantly higher yield of kale (*Brassica oleracea*) as compared to compost application [34].

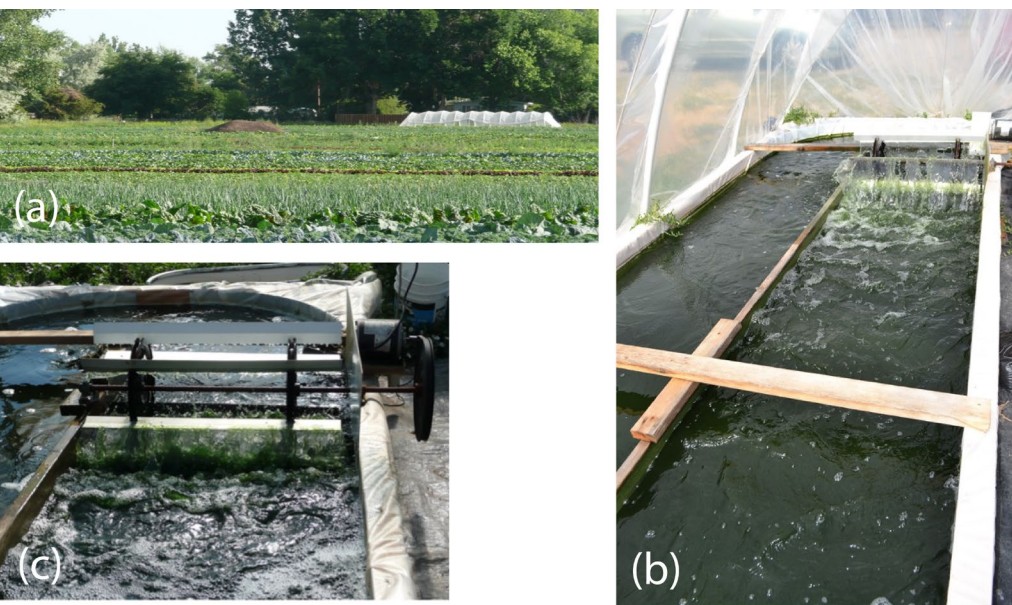

**Figure 1.** Producing cyanobacterial fertilizer on-farm. (**a**) Hoop house in background where cyanobacteria are grown. (**b**) Cyanobacterial production in a raceway inside of a hoop house. (**c**) Paddle wheel used to mix and aerate cyanobacterial culture. Photos by J.G. Davis.

In addition to the reduction in fossil fuel use for N fertilizer production through on-site N fixation powered by photosynthesis, application of cyanobacterial fertilizer through an irrigation system (fertigation) with no dehydration required has also been shown to reduce greenhouse gas ($N_2O$ and $CO_2$) and $NH_3$ emissions [35,36]. These field studies evaluated solid (blood meal, feather meal) and liquid organic fertilizers (cyanobacterial fertilizer, fish emulsion) applied to vegetable crops; solid fertilizers were applied in a pre-plant subsurface band, and liquid fertilizers were applied in small doses throughout the growing season through a surface drip irrigation system. The reduced emissions from cyanobacterial fertilizer were likely due to its application as a liquid through fertigation.

Furthermore, cyanobacteria have been shown to solubilize inorganic phosphates such as hydroxyapatite in agar [37], in submerged conditions such as in rice paddies [38,39], and in incubations of moist soils at field capacity [40,41]. *Anabaena*, *Nostoc*, and *Westiellopsis* sp. have demonstrated the ability to solubilize tricalcium phosphate and hydroxyapatite [29], and several different mechanisms have been proposed [28]. Afkairin et al. [41] demonstrated the effectiveness of *Anabaena* sp. in increasing bioavailability of phosphorus (P) from organic P fertilizers, such as bone meal and rock phosphate.

Cyanobacteria also release extracellular phytohormones including gibberellins, cytokinin, auxin, and abscisic acid [28]. Alvarez et al. [29] thoroughly summarized the scientific literature on the production of phytohormones by cyanobacterial species and plant growth and performance responses to those cyanobacteria. The most commonly reported effects on plants include improving germination, root and shoot length and/or weight, leaf number, or flower parameters. It is difficult to separate the influence of nutrients in cyanobacterial fertilizer from the impact of its accompanying phytohormones on plant growth or physiology. Wenz et al. [42] evaluated whether management of the cyanobacterial production process influenced phytohormone concentrations; they reported that

culture inoculation density was a significant variable in predicting the concentrations of indole-3-acetic acid, indole acetamide, and salicylic acid concentrations. As we learn more, it may become possible to manage the cyanobacterial fertilizer production process to optimize phytohormone levels for different crops.

Cyanobacterial fertilizer has also been reported to result in nutritional benefits. For example, Sukor et al. [43] compared cyanobacterial fertilizer with a no-fertilizer control and three other organic fertilizers applied to lettuce. They reported that cyanobacterial fertilizer increased β-carotene concentration in lettuce leaves in one of the two study years. However, fish emulsion increased β-carotene concentration in both years. The application of indole-3-acetic acid (IAA, a plant growth hormone) was highest in the fish emulsion treatment, and IAA application in the fertilizers was positively correlated with the β-carotene concentration in the lettuce [43].

Zinc (Zn) and iron (Fe) are important minerals in human health, and their concentrations in plants have shown variable responses to cyanobacterial fertilizer application. Both Zn and Fe are usually included in the nutrient solutions commonly used to grow cyanobacteria, and so might end up in the fertilizer that is eventually applied. In a greenhouse study evaluating both liquid and solid cyanobacterial fertilizers, available Zn and Fe concentrations in soil were significantly increased by both fertilizer types, and leaf Zn and Fe were increased in response to solid cyanobacterial fertilizer in kale, pepper, and maize, while the liquid fertilizer increased leaf Zn and Fe concentrations in pepper and maize only [32]. In field studies, the results have been less conclusive. Yoder and Davis [44] reported no significant effect of cyanobacterial fertilizer on Zn and Fe concentrations in kale. Sterle et al. [45] evaluated cyanobacterial fertilizer on peach trees and found that the leaf chlorophyll concentration (measured nondestructively with a soil plant analysis development meter) was higher in trees receiving cyanobacterial fertilizer. Chlorophyll was also positively correlated with leaf Fe concentration [45]. In field studies of lettuce and sweet corn, an interaction was observed between inorganic N sources ($NO_3^-$ and $NH_4^+$) in organic fertilizers and plant Fe concentrations; this interaction complicated the interpretation of the results [46]. Nonetheless, in some cases, cyanobacterial fertilizer applications have resulted in higher Zn and/or Fe concentrations.

Plant water use efficiency (WUE) can also be influenced by fertilizer type. Field WUE (fWUE) is defined as crop yield divided by water used, while instantaneous WUE (iWUE) is defined as the ratio of net photosynthetic rate to transpiration rate. Sukor et al. [47] compared cyanobacterial fertilizer to other organic fertilizers (fish emulsion, blood meal, and feather meal) applied to sweet corn and found that cyanobacterial fertilizer had the highest yield and the highest WUE. In an effort to understand the mechanism behind the impact of fertilizer type on the WUE, these investigators correlated the amounts of plant growth hormones applied with the fertilizers and found that the amount of salicylic acid applied was positively correlated with both fWUE and iWUE. The cyanobacterial fertilizer had the highest amount of salicylic acid applied, an order of magnitude higher than the other fertilizers evaluated [47]. Cyanobacterial fertilizer might therefore improve WUE, potentially through the application of salicylic acid in addition to N.

Several studies have recorded shifts in the soil microbial population following cyanobacterial applications [29]. For example, Rogers and Burns [48] reported increases in some bacterial groups, actinomycetes, and fungi in a greenhouse study after inoculating a soil with *Nostoc muscorum* [48]. Afkairin [49] evaluated *Anabaena cylindrica* applications in two organic cropping systems (an annual vegetable rotation and an untilled peach orchard) grown on different soil types. *A. cylindrica* cyanobacterial fertilizer significantly impacted the biomass and structure of soil microbial communities in both cropping systems. The bacteria and actinomycete populations increased in the vegetable system, and the soil fungi and arbuscular mycorrhizal fungi increased in the peach orchard. Afkairin [49] also demonstrated that cyanobacterial fertilizers can affect soil microbial communities, even in short periods of time (<1 year).

In addition to shifts in microbial communities, cyanobacteria can also have antagonistic effects against plant diseases through their production of bioactive compounds with antibacterial, antifungal, and antialgal potential [28]. Singh et al. [28] and Alvarez et al. [29] summarized the extant literature regarding cyanobacterial production of these compounds, but it is clear that more work remains to be carried out to be able to harness cyanobacteria as natural antidisease agents. In addition to biotic stresses, cyanobacteria have also been reported to aid plants facing abiotic stresses, such as salinity, drought, and heavy metal pollution [50]. Most research on both biotic and abiotic stresses has been carried out in laboratory environments to date, with little or no information available from field-scale work.

## 4. Challenges for Future Research

There are a number of trade-offs to be considered regarding cyanobacterial biofertilizer production in the future. For example, some scientists have chosen to work on low-tech systems to simplify cyanobacteria production for on-farm systems (Figure 2). However, such low-tech systems, which are usually open systems, are more vulnerable to contamination of the cyanobacterial culture with other microorganisms. In addition, low-tech systems generally have slower growth rates than more advanced systems and would therefore require a greater land base than a high-tech system. This is a serious limitation because land is often the primary resource of smallholder farmers, especially in developing countries. On the other hand, low-tech on-farm systems can often make use of locally available materials to minimize dependence on imported materials and keep expenses low [31,51,52]. In general, more advanced cyanobacteria production systems are more productive but also more expensive. In addition, high-tech systems usually require a higher level of management and training and might have increased costs for operations and maintenance.

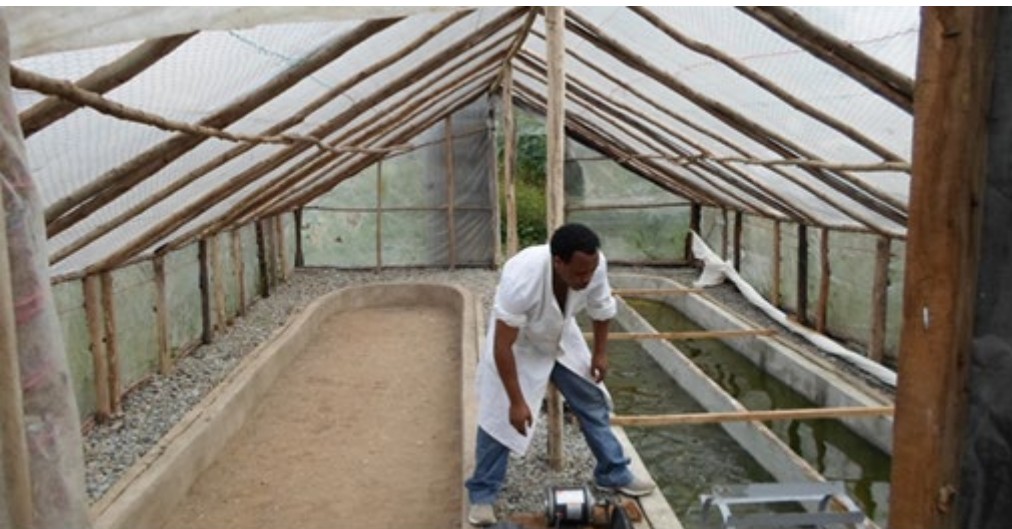

**Figure 2.** Prototype cyanobacteria raceways in production in Ethiopia, created using local materials and knowledge. The cyanobacteria were cultured from local soils. Photo by J.G. Davis. S. Yigrem pictured.

Chemical challenges also exist to maintain a favorable growth environment for cyanobacterial fertilizer cultures. Even simply sourcing nutrients for cyanobacterial fertilizer cultivation might be a challenge in some parts of the world. In addition, the chemistry of the nutrients themselves can limit availability for cyanobacterial fertilizer; for example, P solubility can be low at the neutral to slightly alkaline pH optimal for cyanobacteria growth. Stability of the pH can itself become an issue if high rates of photosynthesis remove pH-buffering alkalinity from the water in which the cyanobacteria grow. After the removal of residual alkalinity, the pH can increase to extremely alkaline levels, resulting in the

loss of the cyanobacterial culture. These challenges will need to be met or managed if widespread production of cyanobacterial fertilizer is to be realized.

The production system and level of technology utilized to grow cyanobacteria depend on the value of the product and the level of control needed over the production process [53]. Open systems that use sunlight have disadvantages such as contamination by other microorganisms or algae, or predation by grazers (including microbiological predators). Additionally, open systems are more vulnerable to diurnal fluctuation of environmental parameters. In contrast, closed systems that utilize sunlight are less susceptible to contamination and can achieve higher cell densities while also taking advantage of solar radiation; however, the cost of these systems is much higher due to the need for transparent materials, gas management (i.e., oxygen and $CO_2$), and temperature maintenance [53]. Finally, closed systems that utilize artificial light as a source of energy can also be used to cultivate cyanobacteria. These systems provide real-time control and optimization of parameters, but the cost increases significantly. On the other hand, large amounts of high-quality cyanobacterial biomass can be produced, and since escape to the environment is improbable, cultivation of genetically modified cyanobacteria is possible [53]. When cyanobacterial cultures are of high value and purity is essential, then these high-tech systems may be worth the additional cost; however, when cyanobacteria are being grown to use as fertilizer, the value is not perceived to be as high, and purity is not as necessary. Therefore, systems that use natural light are preferable in most cases for growing cyanobacteria as N fertilizer.

While an on-farm production model for cyanobacterial fertilizer has a number of advantages, another option could be a distributed network of "fertilizer farms" where cyanobacterial biofertilizer is grown for sale to local farmers. This approach would allow a greater technological investment and a higher level of management, while minimizing fossil fuels used in transportation. Additionally, since cyanobacterial N fertilizer production is not dependent on supplies of natural gas, as $NH_3$ production with the Haber–Bosch process can be, cyanobacteria can potentially be grown without the infrastructure limitations associated with conventional N fertilizer production.

There are a variety of options for sourcing cyanobacterial cultures. In order to prevent environmental contamination in case of a breach in a raceway or pond, many scientists have chosen to culture local cyanobacteria, such as *Anabaena cylindrica* [30] or *Anabaena oscillariodes* [31], from local soil or water sources [30,31,51]. Others have chosen marine cyanobacteria because marine cyanobacteria have fewer potential contaminants. Although some cyanobacterial species can be grown in seawater, the application of these cyanobacteria as fertilizer for terrestrial plants could result in toxic effects on plants due to excessive salinity. The utility of cyanobacteria for various purposes can be enhanced through genetic manipulation or modification which can potentially increase growth, photosynthetic efficiency, biomass yield, or other important parameters [28]; the potential for environmental contamination by genetically modified organisms (GMO) can be minimized through the use of closed bioreactors. However, many farmers avoid growing or utilizing GMOs, especially farmers in countries with GMO regulations or those following organic standards that prohibit the use of GMOs. From a biosecurity perspective, the use of locally cultured cyanobacteria is probably the safest option, but locally cultured cyanobacteria might not be the most productive in terms of N fixation or production of plant growth promoting substances.

Cyanotoxins are another potential risk of cyanobacterial fertilizer use. Reports of cyanobacterial blooms and accompanying cyanotoxins have grown in frequency in recent years [54,55]. Cyanotoxins pose threats to both human and animal health, including livestock and wildlife, usually through exposure to toxin-containing water. The most common cyanotoxins include microcystins, cylindrospermopsin, and anatoxins [55]. Concentrations of all cyanotoxin groups generally increase with increasing concentrations of N and P in the water body [56]. It is critically important that cyanobacteria used as fertilizer do not contain toxins, since there is a chance of cyanotoxins being introduced into the food chain through

this pathway. One way to reduce the risk of cyanotoxins is to produce cyanobacterial biofertilizers in water with low levels of N and P; conveniently, biological N fixation of $N_2$ from the atmosphere is also maximized when N levels in the water are low. On the other hand, efforts have been made to recycle nutrients from industrial wastewaters to grow cyanobacteria for use as a fertilizer or soil amendment [57]; the use of nutrient-rich wastewaters for cyanobacterial production might enhance the risk from cyanotoxins, due to the increased potential for cyanotoxin production. To ensure that no toxins are present, it is good practice to analyze cyanobacterial fertilizers for cyanotoxins prior to their use [42]. Wenz et al. [42] used mass spectroscopy to analyze their *Anabaena* sp. cyanobacterial biofertilizer for anatoxin-a (LOD > 0.06 µg $L^{-1}$), microcystins (LOD > 0.05 µg $L^{-1}$), and cylindrospermopsin (LOD > 0.1 µg $L^{-1}$); none of these cyanotoxins were detected.

Economic feasibility is essential for the adoption and long-term success of cyanobacterial fertilizers. Start-up costs include capital costs such as land, water, construction of raceways and aerators or mixers, and power to operate the aerators and mixers. Investments required in capital and operational costs are very high and necessitate efforts to reduce production costs to achieve profitability [53]. An economic evaluation of the production of microalgal biomass as an energy source (not on cyanobacterial fertilizer) reported that of the scenarios they evaluated, the only potentially profitable scenario was the extraction and commercialization of concentrated proteins [58]. Unfortunately, no economic evaluation of cyanobacterial fertilizer compared with urea is available in the scientific literature.

For operation of the production system, cyanobacterial cultures, nutrients to feed the cultures (e.g., P and Fe), maintenance costs, and trained personnel to manage the system are all important. It is essential to minimize the opportunity cost of the land set aside for cyanobacterial biofertilizer production by siting the production units either on nonarable land or on land with low agricultural productivity. When these costs have been defined for a specific locale, then the time required to break even can also be calculated so that an investor can plan accordingly. Another vital aspect to long-term economic success will be a market analysis and distribution plan, so that there are enough farmers in close proximity to the production units to ensure profitability.

As cyanobacterial fertilizer production systems become optimized, quantification of the carbon footprint and economic costs and benefits of this fertilizer system will be critical to full-scale adoption. To become a successful replacement for urea and other Haber–Bosch-based fertilizers, cyanobacterial fertilizers will need to outcompete urea both in the marketplace and in favorable environmental outcomes. Whether this eventuality comes to pass remains to be seen.

## 5. Conclusions

In light of the many disadvantages of N fertilizers produced through the Haber–Bosch process, the potential benefits of cyanobacterial fertilizer use are worthy of further investigation and investment. Better plant growth, increased crop yields, more efficient nutrient and water use, decreased geopolitical risks, reduced emissions from fertilizer distribution and transport, lower greenhouse gas emissions following field application, and more sustainable food production are some of the potential benefits that might be harvested from the intensive cultivation of cyanobacteria as N fertilizer. Remaining challenges include the optimization of the cyanobacterial fertilizer production system for different environments and precise economic analyses to optimize its competitiveness around the world.

**Author Contributions:** Conceptualization, M.S.M. and J.G.D.; writing—original draft preparation, J.G.D. and M.S.M.; writing—review and editing, M.S.M. and J.G.D. All authors have read and agreed to the published version of the manuscript.

**Funding:** This research received no external funding.

**Data Availability Statement:** No new data were collected or created.

**Conflicts of Interest:** The authors declare no conflict of interest.

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
