# Peer review of "Beyond Soil Inoculation: Cyanobacteria as a Fertilizer Replacement"

_nitrogen, doi:10.3390/nitrogen4030018_

Round 1

Reviewer 1 Report

The work is well done and relatively comprehensive. Nevertheless, it needs two fundamental issues that need to be worked out.

1) Economic efficiency. It is mentioned only marginally and generally in the article. In reality, however, this is a crucial aspect of the future use of this "fertilizer". Economic efficiency will be different everywhere in the world, and it is impossible to describe it precisely, but you should try to work on this issue.

2) The principle of nitrogen uptake by plants. Please describe what forms of nitrogen contain cyanobacteria, how they are released, etc. From the point of view of plant nutrition, this is essential. Mineral nitrogen fertilizers also have different effects (slow or fast release), and their application is chosen accordingly.

Author Response

Thank you for taking the time to provide us with a review. We respond to your comments below in italics.

The work is well done and relatively comprehensive. Nevertheless, it needs two fundamental issues that need to be worked out.

1) Economic efficiency. It is mentioned only marginally and generally in the article. In reality, however, this is a crucial aspect of the future use of this "fertilizer". Economic efficiency will be different everywhere in the world, and it is impossible to describe it precisely, but you should try to work on this issue.—We have added a paragraph on economic feasibility in the Challenges for Future Research section of the manuscript (see lines 357-369).

2) The principle of nitrogen uptake by plants. Please describe what forms of nitrogen contain cyanobacteria, how they are released, etc. From the point of view of plant nutrition, this is essential. Mineral nitrogen fertilizers also have different effects (slow or fast release), and their application is chosen accordingly.—We have added a few sentences to address this in the Potential Benefits of Cyanobacterial Fertilizer section (see lines 158-164).

Reviewer 2 Report

Dear authors,

Thank you for the opportunity to review this Manuscript (Beyond Soil Inoculation: Cyanobacteria as a Fertilizer Replacement). The study brings interests results about production of cyanobacterial fertilizer. Howerver, there is no data and no information about economic viability and use of cyanobacterial as fertilizer. I suggest the authors improve the results of the study. 

Author Response

Thank you for providing us with a helpful review. We have responded to your comments below.

Thank you for the opportunity to review this Manuscript (Beyond Soil Inoculation: Cyanobacteria as a Fertilizer Replacement). The study brings interests results about production of cyanobacterial fertilizer. Howerver, there is no data and no information about economic viability and use of cyanobacterial as fertilizer. I suggest the authors improve the results of the study. – We have added a paragraph on economic feasibility in the Challenges for Future Research section of the manuscript (see lines 357-369).

Reviewer 3 Report

The manuscript with the title “Beyond Soil Inoculation: Cyanobacteria as a Fertilizer Replacement” explores the potential of cyanobacteria to become a veritable N fertilizer.

The manuscript is well written and connected to the current body of literature.

I suggest a short Introduction paragraph before going into the chapters of the paper. This introduction paragraph should provide general information and background for the general public on topic such as: importance of N for crops, and enumerate sources of N. Current challenges that would make prospecting for new N sources relevant. Name clear aim and objectives of this review.

I advise authors to provide more specific information (particularly the microbiology taxonomy aspects and delimitation).  

Chapter 2

I advise authors to add a brief delimitation on Free-living and Symbiotic bacteria that can fix N, with emphasis on those traditionally used so far in agriculture. Please note that Cyanobacteria are also belonging to Bacteria domain of life and found on soil crust usually dormant. Please also note there are other microorganisms that can enhance N uptake, such as fungi (arbuscular mycorrhiza – phylum Glomeromycota, but also Dark Septate Endophytes – phylum Ascomycota etc.). Inoculants with some of these microorganisms also exist. These are also hot topics and perhaps shall be mentioned here.

Chapter 3 debuts with “Cyanobacteria can …”, without briefly explaining in a sentence that these are microorganisms classified as …, belonging to phylum … etc., having high diversity but only a few genera have been studied so far, therefore remaining unexplored for the most part…

Conclusions should mirror the objectives of the review that should be proposed at the end of the brief introduction.

Best regards.

Author Response

Thank you for your thorough review. We have responded to your comments below in italics.

The manuscript with the title “Beyond Soil Inoculation: Cyanobacteria as a Fertilizer Replacement” explores the potential of cyanobacteria to become a veritable N fertilizer.

The manuscript is well written and connected to the current body of literature.

I suggest a short Introduction paragraph before going into the chapters of the paper. This introduction paragraph should provide general information and background for the general public on topic such as: importance of N for crops, and enumerate sources of N. Current challenges that would make prospecting for new N sources relevant. – We have added a paragraph as suggested. See lines 27-35.

Name clear aim and objectives of this review. – We have added objectives of the review to lines 64-67.

I advise authors to provide more specific information (particularly the microbiology taxonomy aspects and delimitation). – We have added more specific information regarding phylogeny or taxonomy of the cultures used where that information is available. For example, Wolde et al. (2020) used Anabaena oscillariodes, and Barminski et al. (2016) used Anabaena cylindrica, whereas other studies focused on other species (e.g., Obana et al. [2007] used Nostoc sp.).

 Chapter 2

I advise authors to add a brief delimitation on Free-living and Symbiotic bacteria that can fix N, with emphasis on those traditionally used so far in agriculture. Please note that Cyanobacteria are also belonging to Bacteria domain of life and found on soil crust usually dormant. – This has chiefly been covered in section 1 (introduction); we have added additional information and citations regarding cyanobacteria species and soil crusts (lines 45-46), particularly for soil restoration (e.g., Chamizo et al, 2018; Roncero-Ramos et al., 2019).

Please also note there are other microorganisms that can enhance N uptake, such as fungi (arbuscular mycorrhiza – phylum Glomeromycota, but also Dark Septate Endophytes – phylum Ascomycota etc.). Inoculants with some of these microorganisms also exist. These are also hot topics and perhaps shall be mentioned here. – We have added a sentence to section 1 (introduction) discussing fungal inocula in addition to bacterial inocula (lines 53-56).

Chapter 3 debuts with “Cyanobacteria can …”, without briefly explaining in a sentence that these are microorganisms classified as …, belonging to phylum … etc., having high diversity but only a few genera have been studied so far, therefore remaining unexplored for the most part… – We have added an introductory sentence to section 3 (lines 121-124).

Conclusions should mirror the objectives of the review that should be proposed at the end of the brief introduction. – We have added conclusions to mirror the objectives in the Conclusions section (see lines 377-378 and 383-386).

Round 2

Reviewer 1 Report

Dear authors,

You have worked on my comments but should have done it more precisely. Please include at least an estimate of the economic comparison of cyanobacteria and conventional nitrogen fertilizers.

Author Response

Dear Reviewer,

We agree with you that economic analysis is key to the adoption of cyanobacterial fertilizer. As this is a review article, we have searched the literature and cannot find an economic analysis of cyanobacterial fertilizer as compared to urea (or conventional fertilizers in general) in the scientific literature. In addition, since we (the authors) are not economists, it would be irresponsible of us to guess at what an economic comparison might conclude. 

We did find a paper that reported an economic evaluation of microalgal biomass for energy production (Bravo-Fritz, C.P.; Sáez-Navarrete, C.A.; Herrera-Zeppelin, L.A.; Varas-Concha, F. Multi-scenario energy-economic evaluation for a biorefinery based on microalgae biomass with application of anaerobic digestion, Algal Research 2016, 16, 292-307. https://doi.org/10.1016/j.algal.2016.03.028.), and we have added this paper to the References (#58) and included its conclusions in the addition we made to the paper in lines 360-366 (highlighted in yellow in the manuscript to avoid confusion with our previous edits):

“Investments required in capital and operational costs are very high and necessitate efforts to reduce production costs to achieve profitability [53]. An economic evaluation of the production of microalgal biomass as an energy source (not specifically on cyanobacterial fertilizer) reported that the only potentially profitable scenario was the extraction and commercialization of concentrated proteins [58]. Unfortunately, no economic evaluation of cyanobacterial fertilizer compared with urea is available in the scientific literature.”

Thank you for your review.

Reviewer 2 Report

There is no new information or data in the manuscript. 

Author Response

Dear Reviewer,

Thank you for taking the time to review our manuscript. You are correct that since this is a review article, there is no new data in the manuscript. 

Since your initial review focused on the economic analysis, we have made an effort to provide additional information to that section. As this is a review article, we have searched the literature and cannot find an economic analysis of cyanobacterial fertilizer as compared to urea (or conventional fertilizers in general) in the scientific literature. 

We found a paper that reported an economic evaluation of microalgal biomass for energy production (Bravo-Fritz, C.P.; Sáez-Navarrete, C.A.; Herrera-Zeppelin, L.A.; Varas-Concha, F. Multi-scenario energy-economic evaluation for a biorefinery based on microalgae biomass with application of anaerobic digestion, Algal Research 2016, 16, 292-307. https://doi.org/10.1016/j.algal.2016.03.028.), and we have added this paper to the References (#58) and included its conclusions in the addition we made to the paper in lines 360-366 (highlighted in yellow in the manuscript to avoid confusion with our previous edits):

“Investments required in capital and operational costs are very high and necessitate efforts to reduce production costs to achieve profitability [53]. An economic evaluation of the production of microalgal biomass as an energy source (not specifically on cyanobacterial fertilizer) reported that the only potentially profitable scenario was the extraction and commercialization of concentrated proteins [58]. Unfortunately, no economic evaluation of cyanobacterial fertilizer compared with urea is available in the scientific literature.”

Thank you for your review.